# Four families of folate-independent methionine synthases

**Morgan N. Price**[ID][1]*, **Adam M. Deutschbauer**[ID][1], **Adam P. Arkin**[ID][1,2]*

**1** Environmental Genomics and Systems Biology, Lawrence Berkeley National Lab, Berkeley, California, United States of America, **2** Department of Bioengineering, University of California, Berkeley, California, United States of America

* morgannprice@yahoo.com (MNP); aparkin@lbl.gov (APA)

**Data Availability Statement:** The fitness data are archived in FigShare (https://doi.org/10.6084/m9.figshare.13146419.v1) and are also available in the Fitness Browser (http://fit.genomics.lbl.gov/).

## Abstract

Although most organisms synthesize methionine from homocysteine and methyl folates, some have "core" methionine synthases that lack folate-binding domains and use other methyl donors. *In vitro*, the characterized core synthases use methylcobalamin as a methyl donor, but *in vivo*, they probably rely on corrinoid (vitamin B12-binding) proteins. We identified four families of core methionine synthases that are distantly related to each other (under 30% pairwise amino acid identity). From the characterized enzymes, we identified the families MesA, which is found in methanogens, and MesB, which is found in anaerobic bacteria and archaea with the Wood-Ljungdahl pathway. A third uncharacterized family, MesC, is found in anaerobic archaea that have the Wood-Ljungdahl pathway and lack known forms of methionine synthase. We predict that most members of the MesB and MesC families accept methyl groups from the iron-sulfur corrinoid protein of that pathway. The fourth family, MesD, is found only in aerobic bacteria. Using transposon mutants and complementation, we show that MesD does not require 5-methyltetrahydrofolate or cobalamin. Instead, MesD requires an uncharacterized protein family (DUF1852) and oxygen for activity.

## Author summary

Methionine is one of the amino acids that make up proteins, and the final step in methionine synthesis is the transfer of a methyl group. In most organisms, the methyl group is obtained from methyl folates, but some anaerobic bacteria and archaea are thought to use corrinoid (vitamin B12-binding) proteins instead. By analyzing the sequences of the potential methionine synthases across the genomes of diverse bacteria and archaea, we identified four families of folate-independent methionine synthases. For three of these families, we can use co-occurrence with corrinoid proteins to predict their likely partners. We show that the fourth family does not require vitamin B12; instead, it obtains methyl groups from an oxygen-dependent partner protein. Our results will help us understand the growth requirements of diverse bacteria and archaea.

**Funding:** This work was funded by ENIGMA, a Scientific Focus Area Program at Lawrence Berkeley National Laboratory, supported by the U. S. Department of Energy, Office of Science, Office of Biological and Environmental Research (https://www.energy.gov/science/ber/biological-and-environmental-research) under contract DE-AC02-05CH11231 and granted to AMD and APA. The funders had no role in study design, data collection and analysis, decision to publish, or preparation of the manuscript.

**Competing interests:** The authors have declared that no competing interests exist.

## Introduction

Methionine is required for protein synthesis and is also a precursor to S-adenosylmethionine, which is the methyl donor for most methyltransferases and is required for polyamine biosynthesis. Methionine is synthesized from aspartate by reduction and sulfhydrylation to homocysteine, and then the transfer of a methyl group to homocysteine to give methionine (Fig 1). There are two well-studied forms of methionine synthase, both of which obtain the methyl group from 5-methyltetrahydrofolates (5-methyl-THF). MetH requires cobalamin (vitamin B12) or other cobamides as a cofactor, while MetE does not require any organic cofactor. For the same mass of protein, MetH is about 40 times more active than MetE [1,2]. *Escherichia coli*, which cannot synthesize cobamides, has both enzymes. There are also methyltransferases that convert homocysteine to methionine by using methylated nutrients such as glycine betaine or S-methylmethionine [3]. Here we will focus on the synthesis of methionine without special nutrient requirements.

Besides MetH and MetE, three other types of methionine synthases have been reported. These "core" methionine synthases [4] are homologous to the C-terminal catalytic domain of MetE and do not contain any other domains. In particular, they lack the N-terminal domain of MetE that is involved in binding folate [5]. We describe each of these enzymes below.

First, a core methionine synthase from the methanogen *Methanobacterium thermoautotrophicum* has been studied biochemically [6]. We will call this protein MesA (methionine synthase A; UniProt:METE_METTM). *In vitro*, MesA transfers methyl groups from methylcobalamin to homocysteine to form methionine (Fig 1), but it has no activity with 5-methyl-THF or 5-methyltetrahydromethanopterin as substrates [6]. (Tetrahydromethanopterin is a cofactor for methanogenesis that is similar to THF.) Because MesA has a very weak affinity for methylcobalamin (the Michaelis-Menten constant is above 20 mM), and because most of the cobalamin in methanogens is bound to corrinoid proteins, the physiological substrates of MesA are probably methyl corrinoid proteins [6]. It might seem surprising that MesA accepts methyl groups from cobamides when it is homologous to the cobalamin-independent enzyme MetE, but the catalytic mechanisms of MetE and MetH are similar: both MetE and MetH rely on a zinc cofactor to deprotonate the sulfur atom of homocysteine and activate it as a nucleophile [7].

Second, a core methionine synthase from the bacterium *Dehalococcoides mccartyi* was recently identified [4]. We will call this protein MesB (UniProt:A0A0V8M4G6). MesB uses methylcobalamin, but not 5-methyl-THF, as a substrate *in vitro*, and MesB cannot complement a *metE-* strain of *E. coli* [4]. MesB was proposed to obtain methyl groups from the iron-sulfur corrinoid protein (CoFeSP) of the Wood-Ljungdahl pathway [4].

Third, a genetic study identified an unusual methionine synthase in *Acinetobacter baylyi* ADP1 [8]. ACIAD3523 (UniProt:Q6F6Z8) is required for methionine synthesis in the absence of cobalamin, and so is the adjacent gene ACIAD3524. Although ACIAD3523 was originally annotated as a MetE protein and is so described in the genetic study, ACIAD3523 lacks the N-terminal folate-binding domain, and it is distantly related to the C-terminal catalytic domain of MetE (under 30% identity). The associated protein ACIAD3524 belongs to the uncharacterized family DUF1852 (Pfam PF08908, [9]; DUF is short for domain of unknown function). We will call these proteins MesD and MesX.

Homologs of these core methionine synthases are found in diverse bacteria and archaea, but it is not clear if these homologs have the same functions. And even for the characterized enzymes, the source of the methyl groups are not known. Furthermore, as we will explain, some organisms that grow in minimal media do not contain any of the known forms of methionine synthase.

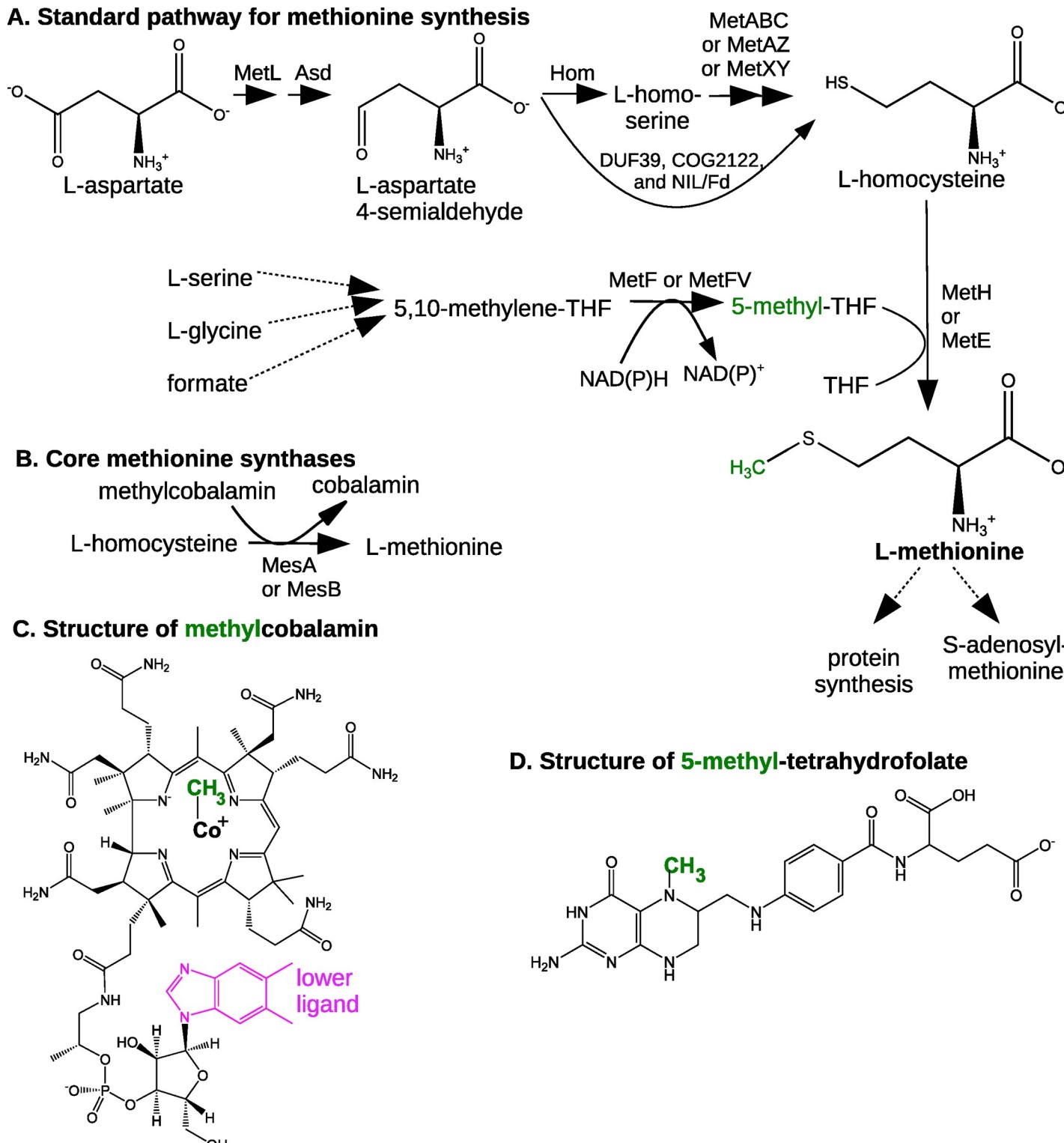

**Fig 1. Overview of methionine synthesis.** (A) The standard pathway with 5-methyl-THF. (B) The reaction catalyzed by the core methionine synthases MesA and MesB. (C) The structure of methylcobalamin. Cobalamin has 5,6-dimethylbenzimidazole as the lower ligand, but many organisms use cobamides with other lower ligands. (D) The structure of 5-methyl-THF. Although THF is shown with a single glutamyl residue (at right), in the cell, THF is usually polyglutamylated.

To address these questions, we analyzed the phylogenetic distribution of core methionine synthases. The three characterized core methionine synthases are distantly related to each other (under 30% pairwise amino acid identity), so we used them to define three families. We also noticed that several genera of anaerobic archaea lack all known forms of methionine synthase, but grow in minimal media. These archaea contain another family of putative core methionine synthases, with conserved functional residues, which we call MesC.

We found that MesA is found solely in methanogens. MesB and MesC are found solely in organisms with the Wood-Ljungdahl pathway, so we propose that most members of both families obtain methyl groups from CoFeSP. MesD is found solely in aerobic bacteria. By using pooled mutant fitness assays and complementation assays, we will show that 5-methyl-THF is not the methyl donor for MesD, that MesD requires both MesX and oxygen for activity, and that MesD and MesX suffice to convert homocysteine to methionine in *E. coli*.

## Results and discussion

### Identification of split MetE proteins and the MesC family

We previously ran the GapMind tool for reconstructing amino acid biosynthesis against 150 genomes of bacteria and archaea that grow in defined minimal media without any amino acids present [10]. After updating our analysis to account for MesB and MesD, six archaeal genomes lacked candidates for any of the known types of methionine synthase (MetH, MetH split into two or three parts [10,11], MetE, MesA, MesB, or MesD).

We searched for additional candidates for methionine synthase by using the profile of protein family COG0620 [12] and PSI-BLAST [13]. COG0620 matches both the N-terminal (folate-binding) and C-terminal (homocysteine-activating) domains of MetE, as well as MesA, MesB and MesD.

In the hyperthermophilic archaea *Pyrolobus fumarii* 1A, we identified two hits to COG0620, which correspond to the two domains of MetE. PYRFU_RS09465 contains the N-terminal domain (Meth_synt_1 in Pfam) and PYRFU_RS01495 contains the C-terminal domain (Meth_synt_2 in Pfam). The homologs of these proteins from *Pyrococcus furiosus* (PF1268 and PF1269, respectively) are encoded adjacent to each other and form a complex [14], which suggests that they comprise a split MetE. Split MetE proteins are found primarily in archaea, and they appear to be the sole form of methionine synthase in most of the thermophilic or halophilic archaea (see Materials and Methods). The two parts of split MetE are encoded adjacent to each other in diverse archaea and in the bacterium *Sulfobacillus sp*. hq2. This confirms that these proteins form a conserved system. Many of the previously-proposed "core" methionine synthases from archaea (see the first figure of [4]) are probably catalytic subunits of split MetE proteins; this includes representatives from the genera *Acidianus*, *Aeropyrum*, *Haloferax*, *Natronomonas*, *Pyrobaculum*, *Pyrococcus*, and *Sulfolobus*. Given the domain content of split MetE proteins, we predict that they use methyl-THF or other methyl pterins as their methyl donors. (Thermophilic archaea are thought to use alternate pterins, instead of tetrahydrofolates, as their one-carbon carriers [15].)

The other five archaea with missing methionine synthases were strict anaerobes: three strains of methanogenic *Methanosarcina*, an iron-reducing *Ferroglobus placidus*, and a sulfite-reducing *Archaeoglobus veneficus*. All genomes contained one or more putative core methionine synthases, such as MA_0053 (UniProt:Q8TUL3). These proteins were similar to each other (33% identity or above) and were more distantly related to MesA, MesB, or MesD (pairwise identities under 30%). We will call them MesC.

## The MesA family is found only in methanogens

Given the four types of core methionine synthases that we are aware of, we searched for likely functional orthologs of each, and we examined their distribution across 7,694 bacteria and 321 archaea from UniProt's reference proteomes [16]. We will discuss MesA first.

Although MesA is distantly related to the other types of core methionine synthases, MesA is similar to split MetE proteins. For instance, the characterized MesA is 38% identical to the putative catalytic component of the split MetE from *Pyrolobus fumarii*. We considered closer homologs to form the MesA family. Using phmmer from the HMMer package (http://hmmer.org/), we found 40 such hits (~40% identity or above). All of these putative MesA proteins were from methanogens: this included representatives of 18 genera from the orders Methanobacteriales, Methanocellales, Methanococcales, Methanomicrobiales, and Methanopyrales.

Although most of the methanogens that have MesA encode the Wood-Ljungdahl pathway, some do not, including several species of *Methanobrevibacter* (such as *M. curvatus*). This suggests that the CoFeSP is not the physiological methyl donor for MesA. Also, some methanogens contain both MesA and MesB, which suggests that the two methionine synthases might use different methyl donors. To identify potential methyl donors for MesA, we examined the protein families that were reported to be universally conserved in methanogens [17]. As far as we know, only two of these families are thought to bind cobamides: Mmp10 (methanogenesis marker protein 10) and MtrA. Mmp10 is a S-adenosylmethionine-dependent methyltransferase [18]; since methionine is the precursor to S-adenosylmethionine, Mmp10 cannot be the source of methyl groups for methionine synthesis. MtrA is the corrinoid subunit of methyltetrahydromethanopterin:coenzyme M methyltransferase [19], which catalyzes the last methyl transfer reaction before reduction to methane. We predict that MesA obtains methyl groups from MtrA.

## The MesB family is linked to the iron-sulfur corrinoid protein of the Wood-Ljungdahl pathway

To identify likely functional orthologs of MesB, we began with the hypothesis [4] that MesB obtains methyl groups from the CoFeSP protein of the Wood-Ljungdahl pathway (Fig 2). This hypothesis is consistent with labeling experiments with *Dehalococcoides mccartyi*, which show that the methyl group of methionine is derived from the methyl group of acetate [20]. It also explains how *D. mccartyi* can biosynthesize methionine despite encoding neither methylene-THF reductase (MetF), which is how most bacteria form methyl-THF, nor AcsE, which transfers methyl groups between CoFeSP and methyl-THF.

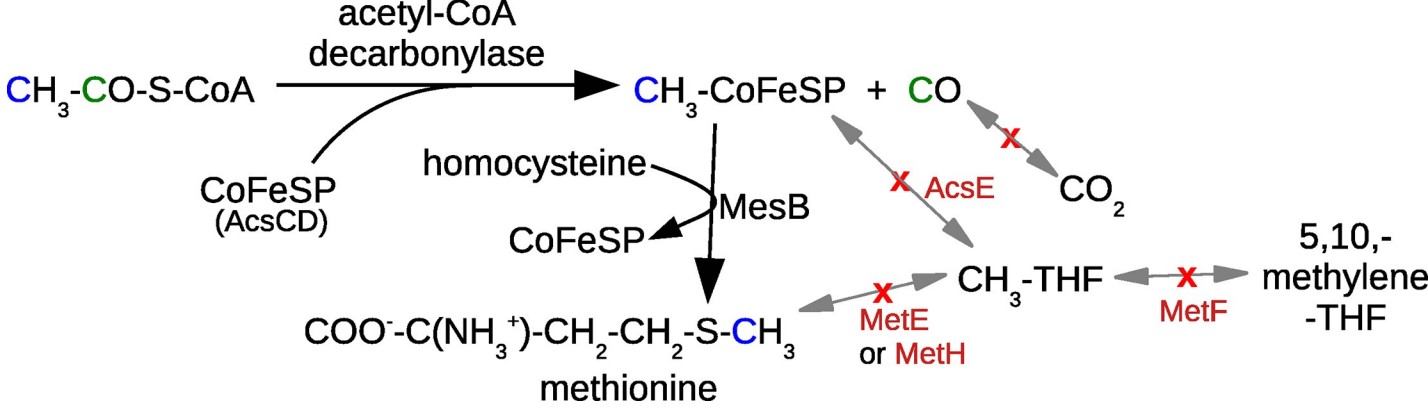

**Fig 2. The proposed pathway for methionine synthesis in *Dehalococcoides mccartyi*.** Steps that are absent from *D. mccartyi* are shown with a red x.

Given this hypothesis, we selected homologs of MesB from MicrobesOnline [21], we built a phylogenetic tree with MUSCLE [22] and FastTree 2 [23], and we checked for the presence or absence of the Wood-Ljungdahl pathway in the organisms that contain close homologs. As shown in Fig 3A, most of the close homologs of MesB are found in organisms with the Wood-Ljungdahl pathway. But there are also close homologs of MesB in some anoxygenic photo-trophic Chloroflexales (*Chloroflexus* or *Roseiflexus*) that lack the Wood-Ljungdahl pathway. As discussed below, the homologs from Chloroflexales lack the residues required for catalysis. We propose that the homologs of MesB in Chloroflexales have another function, while the homo-logs in Wood-Ljungdahl organisms are MesB-type core methionine synthases.

To test more broadly if MesB is present in any organisms that lack the Wood-Ljungdahl pathway, we used phmmer to examine the UniProt reference proteomes. We found 173 hits for MesB with at least 130 bits (~29% identity or above). After excluding the non-enzymatic homo-log from *Roseiflexus* sp. RS-1 and related proteins from two other Chloroflexi, we had 170 puta-tive MesB proteins from 123 proteomes. All of these proteomes contained AcsC or AcsD, which are the two subunits of CoFeSP, and almost all (120/123) contained both AcsC and AcsD.

Many of the bacteria with putative MesB proteins lack the other types of methionine synthase (Fig 3A). Also, many bacteria that contain MesB as the sole putative methionine synthase are known to grow in minimal media. For instance, among the prototrophic bacteria that we previously analyzed with GapMind [10], MesB appears to be the sole methionine synthase in representatives of the genera *Desulfacinum*, *Desulfallas*, *Desulfarculus*, *Desulfatiba-cillum*, *Desulfatiglans*, *Desulfitobacterium*, *Desulfobacca*, and *Thermodesulforhabdus*. These observations strongly suggest that these homologs of MesB are methionine synthases. In con-trast, more distant homologs of MesB are found in bacteria that encode other methionine synthases, which suggests that these MesB-like proteins might have another function.

Although MesB is primarily found in bacteria, it is also found in some methanogens, many of which contain MesA as well. In some methanogens, the *mesB* genes are in an apparent operon with *acsC* and *acsD*, which encode the two subunits of CoFeSP protein (Fig 3B). This conserved operon suggests a direct functional relationship.

The distribution of *mesB*, as well as the gene neighborhood of *mesB* in methanogens, sug-gest that most of the MesB proteins use the CoFeSP protein from the Wood-Ljungdahl path-way as the methyl donor. Some of the bacterial genomes with *mesB* contain 2 or 3 members of the family (Fig 3A), and some of these paralogs might bind another corrinoid protein.

## The MesC family is found in archaea with the Wood-Ljungdahl pathway

All of the organisms that we initially discovered MesC in are anaerobic archaea that encode the Wood-Ljungdahl pathway. To examine the distribution of MesC in the reference prote-omes, we used phmmer with Q8TUL3 from *Methanosarcina acetivorans* as the query. We found a weak hit of MesC to protein A0A166AZ17 from *Methanobrevibacter curvatus*, which does not contain AcsC or AcsD. A0A166AZ17 is more similar to MesA (METE_METTM; 47% identity instead of 22% identity), so we disregarded this hit. The other hits (38 bits or higher) were all to anaerobic archaea that contain AcsC and AcsD, except for the uncultured archaeal lineage MSBL-1. MesC was found in nine draft genomes of MSBL-1, and AcsC and/or AcsD were found in five of these nine. The Wood-Ljungdahl pathway is reported to be pres-ent in many MSBL-1 genomes [24]. Because these are incomplete single-cell genomes, we are not sure if the Wood-Ljungdahl pathway is truly absent from some of the MSBL-1 genomes that encode MesC-like proteins.

Overall, MesC is found in archaea with the Wood-Ljungdahl pathway. This includes metha-nogens from the orders Methanosarcinales and Methanotrichales, iron-reducing *Ferroglobus*

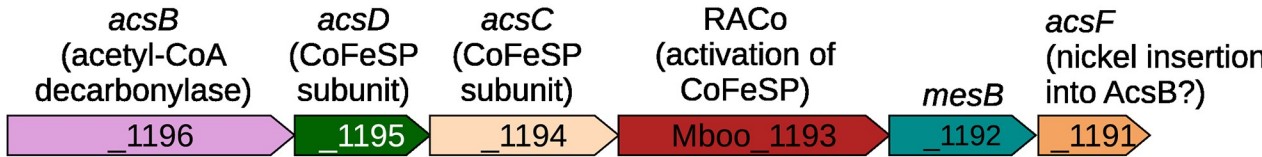

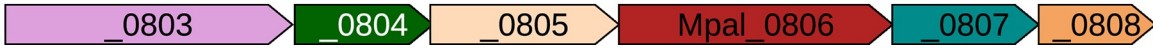

**Fig 3. Comparative genomics links MesB to the Wood-Ljungdahl pathway.** (A) A phylogenetic tree of MesB and related proteins. The MesB family is highlighted in green and a subfamily that lacks Zn-coordinating residues is highlighted in red. On the right, filled symbols indicate the presence in that genome of other methionine synthases or of the Wood-Ljungdahl pathway (*acsBCD*, also known as *cdhCED*). If the genome contains more than one *mesB* gene, we show the number. The tree and the genome properties were rendered with iTOL v5 (https://itol.embl.de/). (B) Conserved clustering of *mesB* with genes from the Wood-Ljungdahl pathway. Gene drawings were modified from MicrobesOnline [21].

and *Geoglobus*, and sulfate-reducing or sulfite-reducing *Archaeoglobus*. We predict that most MesC proteins accept methyl groups from CoFeSP.

Although most of these organisms have just one methionine synthase, representatives of the genus *Methanosarcina* have 2–3 copies of MesC (and no other methionine synthases). The multiple MesC proteins within *Methanosarcina* seem to have arisen by lineage-specific duplications: they cluster together in a phylogenetic tree, and two of the three paralogs are near each other in the genome. *Methanosarcina* can grow on many different methylated compounds via an array of specialized corrinoid proteins [25], so we speculate that some paralogs of MesC accept methyl groups from other corrinoid proteins besides CoFeSP.

It's interesting to consider the prevalence of the different types of methionine synthase across the methanogens. The Methanosarcinales have MesC only, while most other orders of methanogens have MesA only (this includes Methanobacteriales, Methanocellales, Methanococcales, most Methanomicrobiales, and Methanopyrales; S1 Table). Some Methanomicrobiales have both MesA and MesB. The Methanomassiliicoccales have MesB only; because the Methanomassiliicoccales have the Wood-Ljungdahl pathway and lack methyltetrahydromethanopterin:coenzyme M methyltransferase [26], this is consistent with our predictions. The extremely halophilic methyl-reducing methanogens *Methanonatronarchaeum thermophilum* and *Methanohalarchaeum thermophilum* [27] contain split MetE proteins but not other types of methionine synthases. We did not identify MetH or MetE in any methanogens, possibly because most methanogens lack 5-methyl-THF (although 5-methyl-THF might be present in *Methanosarcina* [28]).

## The MesD family is found in diverse aerobic bacteria

To examine the distribution of the core methionine synthase MesD and the associated protein MetX (DUF1852), we first used MicrobesOnline to identify homologs of ACIAD3523 (MesD) that are likely to have the same function in methionine synthesis. Specifically, we used the MicrobesOnline tree-browser to check if they were adjacent to DUF1852. We chose a BLASTp bit score threshold of 390 bits (~55% identity), as homologs above this threshold were almost always adjacent to DUF1852. We also excluded a homolog from the oomycete *Phytophthora capsici*, which could be contamination or might indicate the acquisition of DNA from a *Stenotrophomonas* bacterium. This left 106 genomes from 44 genera that contain MesD. We used FAPROTAX [29] and web searches to check the lifestyles of these genera and found that all 44 of them were aerobic. These include both strict aerobes and facultative anaerobes.

To check more broadly that MesX is found only in aerobic bacteria, we used AnnoTree [30] to obtain a list of 235 genera that contain the corresponding Pfam (PF08908). After removing suffixes (i.e., converting "Erythrobacter_B" to "Erythrobacter"), we found MesX in 206 named genera, of which 170 were distinct from the genus names in MicrobesOnline. We examined a random sample of 50 of these 170 genera and all were aerobic.

Finally, we ran phmmer against UniProt reference proteomes with ACIAD3523 as the query. We considered homologs with a score of at least 560 bits (~72% identity) to be MesD proteins. Almost all of the proteomes with putative MesD proteins (395/399) contained MesX (DUF1852) as well. Organisms with both MesD and MesX included representatives of the α-Proteobacteria, β-Proteobacteria, γ-Proteobacteria, Actinobacteria, Bacteroidetes, and Verrucomicrobia (S1 Table). Genomes with MesD/MesX often contain MetH (75% of cases) or MetE (32% of cases) or both (29% of cases), and MesD/MesX appears to be the sole methionine synthase in just 22% of the organisms that have it. We speculate that MesD/MesX co-occurs with the other methionine synthases because MesD/MesX is only active under aerobic conditions. On the other hand, MesD/MesX has the advantage of not requiring cobalamin (or other cobamides) for activity.

## Conserved functional residues of core methionine synthases

To check if the MesA, MesB, MesC, and MesD families have conserved functions, we examined functional residues involved in catalysis and in binding homocysteine. First, MetE activates homocysteine via a zinc thiolate intermediate, and the residues that coordinate the zinc atom in *E. coli*'s MetE are His641, Cys643, Glu665, and Cys726 [31]. The characterized MesB protein [4] has an aspartate instead of Glu665; this substitution appears to be compatible with zinc binding and catalytic activity.

The zinc-binding residues are highly conserved in the MesA, MesB, and MesD families (Fig 4). The only exceptions are two MesB proteins that have an asparagine instead of Cys726,

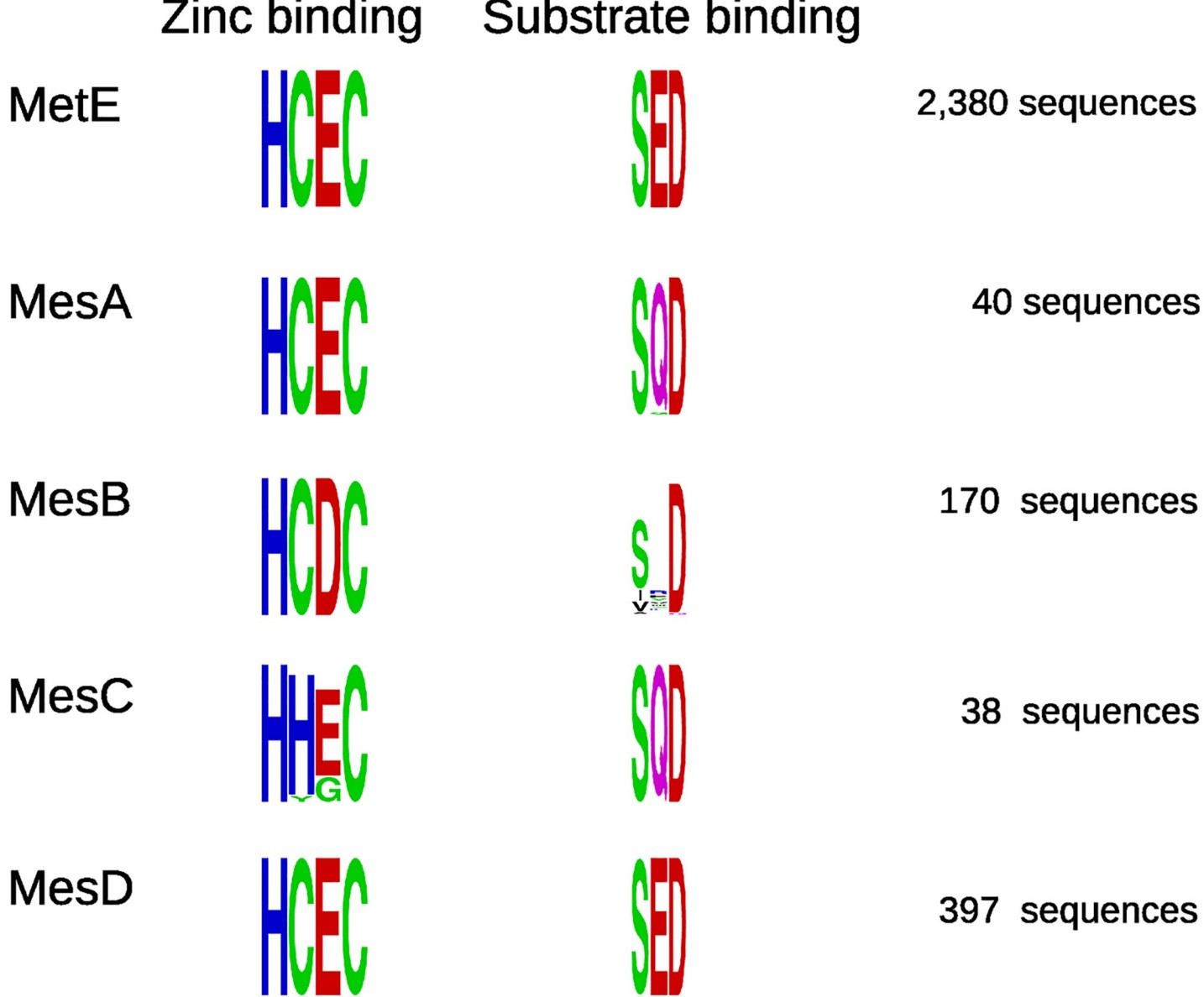

**Fig 4. Functional residues of MetE and of core methionine synthases.** We show sequence logos [33] for the zinc-coordinating and substrate-binding residues of each family of methionine synthases. The height of each position shows its conservation within the family, as measured by information content or bits. In MetE from *E. coli*, the zinc-coordinating residues are H641, C643, E665, and C726, and the substrate-binding residues are S433, E484, and D599.

which might still be compatible with zinc binding: a few zinc-dependent proteins use asparagine as a coordinating residue [32]. In the MesB-like proteins from Chloroflexales (shown in red in Fig 3A, and not included in Fig 4), the zinc-binding residues are not conserved: instead of HCEC or HCDC, they have FSHG, YCDQ, or YREQ.

Most MesC proteins have a histidine instead of Cys643 (Fig 4), which is likely to be compatible with zinc binding. Two MesC proteins from *Methanosarcina* have a tyrosine aligning to Cys643, which might be compatible with zinc binding: a few zinc-dependent proteins use tyrosine as a coordinating residue [32]. Also, these genomes include other representatives of MesC, and those other proteins do have histidines aligning to Cys643. All eight MesC proteins from the uncultured lineage MSBL-1 had a glycine aligning to Glu665, which we would not expect to be compatible with zinc binding [32]. The MSBL-1 genomes contain likely split MetE proteins (i.e., AKJ63_00345:AKJ63_00350), so the MesC proteins from this lineage might have a different function.

We then examined the substrate-binding residues. Structural data suggests that several side chains in MetE form hydrogen bonds with the amino or carboxyl groups of homocysteine or methionine (PDB:1U1J; [34]). (In the *E. coli* residue numbering, Ser433 binds the carboxyl group and Glu484 and Asp599 bind the amino group.) Similarly, in a crystal structure for a MesD protein bound to the methionine analog selenomethionine (PDB:3RPD), the corresponding side chains are in proximity to the amino and carboxyl groups of selenomethionine (Ser22, Glu73, and Asp188 in 3RPD). As shown in Fig 4, MesA and MesC have similar residues, but with a glutamine instead of Glu484. Glutamine could also form a hydrogen bond with the amino group of homocysteine, so we predict that MesA and MesC bind homocysteine (or methionine) in the same manner that MetE and MesD do. The identity of these residues between MesA and MesC supports our prediction that MesC is also a methionine synthase. In MesB, Ser433 and Asp599 are mostly conserved, but Glu484 is not. The region corresponding to Glu484 (around Trp69 in the characterized MesB) is quite variable among MesB proteins and is difficult to align to MetE. Overall, we found that the functional residues for binding zinc and homocysteine were conserved in all four families of core methionine synthases.

## MesD requires MesX and oxygen, but not 5-methyl-THF

We then investigated the function of MesD and MesX in more detail. In particular, we wondered what the source of methyl groups is for MesD. If MesD accepts methyl groups from 5-methyl-THF, then the methylene-THF reductase (MetF) would be required for its activity. But we have several pieces of evidence that MetF is not required for MesD's activity.

First, in *Acinetobacter baylyi*, *mesD* and *mesX* are required for growth if neither methionine nor vitamin B12 are available [8]. (*A. baylyi* also has a cobalamin-dependent methionine synthase (MetH), but MetH is probably not active under these conditions because *A. baylyi* cannot synthesize cobalamin.) In a constraint-based metabolic model of *A. baylyi* in which 5-methyl-THF is a precursor to methionine, *metF* is predicted to be essential for growth [35], which illustrates that MetF is the only known path to 5-methyl-THF. Nevertheless, *metF* from *A. baylyi* is not essential for growth in a defined minimal medium with no vitamins [8]. This suggests that MesD/MesX do not require methyl-THF. A caveat is that another protein from *A. baylyi*, ACIAD1783, is distantly related to MetF, but ACIAD1783 lacks the N-terminal part of MetF and probably has another function (see Materials and Methods).

Second, although most of the organisms with MesD also contain MetH or MetE, we did find 85 proteomes in which MesD seems to be the sole methionine synthase. Many of these proteomes (69%) seem to lack MetF. One of them is *Paenarthrobacter aurescens* TC1, which can grow in a defined minimal medium with the herbicide atrazine as the sole source of carbon

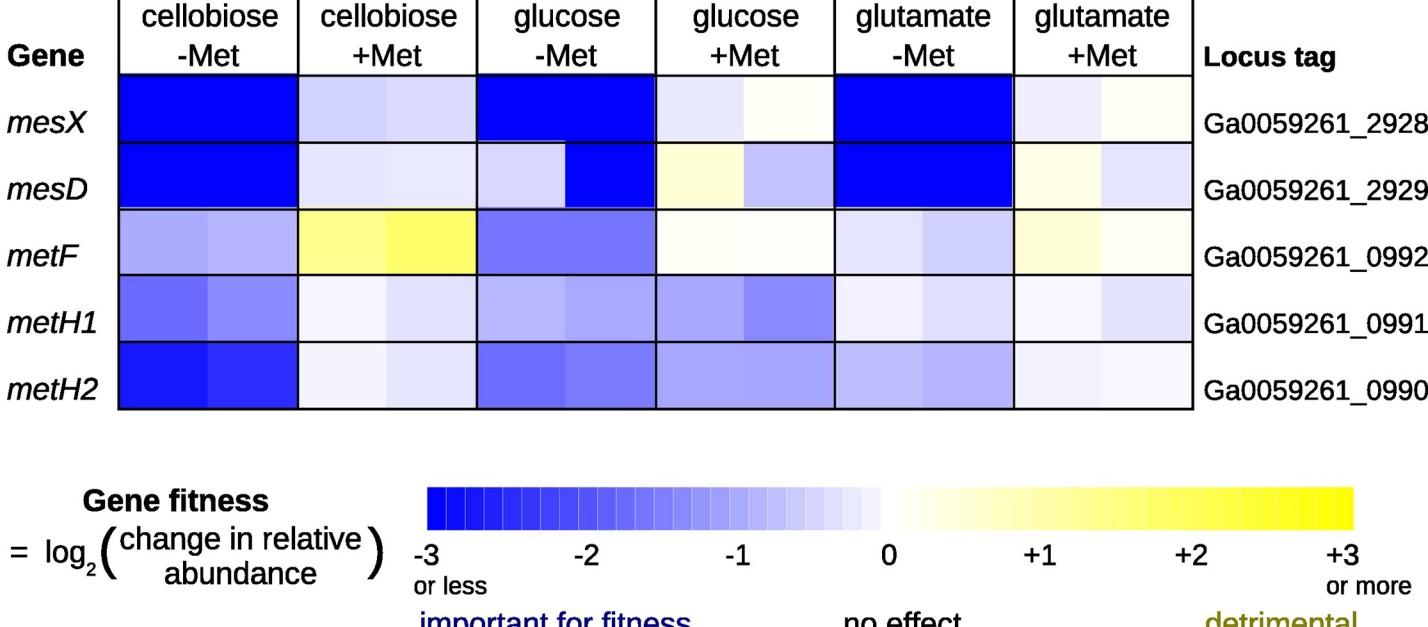

| Gene | cellobiose -Met | cellobiose +Met | glucose -Met | glucose +Met | glutamate -Met | glutamate +Met | Locus tag |
|---|---|---|---|---|---|---|---|
| *mesX* | | | | | | | Ga0059261_2928 |
| *mesD* | | | | | | | Ga0059261_2929 |
| *metF* | | | | | | | Ga0059261_0992 |
| *metH1* | | | | | | | Ga0059261_0991 |
| *metH2* | | | | | | | Ga0059261_0990 |

**Gene fitness**
$$= \log_2 \left( \begin{array}{c} \text{change in relative} \\ \text{abundance} \end{array} \right)$$

-3 or less -2 -1 0 +1 +2 +3 or more

important for fitness no effect detrimental

**Fig 5. *Sphingomonas koreensis* can grow in minimal media by using MesD and not MetF.** A pool of transposon mutants was grown in a defined minimal media with a single carbon source and without added vitamins. Some cultures were supplemented with 250 μM L-methionine. Each cell in the heatmap shows a gene fitness value from a different experiment; each condition has two replicates. A gene fitness value is the log₂ change in the relative abundance of mutants in that gene during that experiment (from inoculation at $OD_{600} = 0.02$ until saturation).

[36]. (This species was formerly known as *Arthrobacter aurescens.)* The degradation pathway for atrazine does not involve 5-methyl-THF or other folate derivatives [37]. Again, it appears that 5-methyl-THF is not required for MesD's activity.

Third, we used high-throughput genetics to investigate methionine biosynthesis in *Sphingomonas koreensis* DSMZ 15582. The genome of *S. koreensis* encodes *mesD*, *mesX*, and *metH* (split into two parts), but not *metE* or genes for the *de novo* biosynthesis of cobamides [38]. If cobalamin is not provided in the media, *mesD* should be required for methionine synthesis. We grew a pool of over 250,000 barcoded transposon mutants of *S. koreensis* [39] in defined media with no cobalamin and with cellobiose, glucose, or glutamate as the sole source of carbon. As shown in Fig 5, both *mesD* and *mesX* were important for fitness unless methionine was added. *MetF* was less important for fitness, especially when glutamate was the carbon source. We observed mild phenotypes for disrupting either part of the split *metH*, which might indicate some carry-over of vitamin B12 from the media used to recover the mutant pool from the freezer. (The recovery media contains tryptone, which is often made by hydrolyzing animal protein.) Besides *mesD* and *mesX*, most of the genes whose mutants were rescued by added methionine were involved in homocysteine biosynthesis or other metabolic processes that are not necessary if methionine is available (S1 Fig). The few other genes whose mutants were rescued are not expected to be involved in methionine biosynthesis (S1 Fig). Overall, in *S. koreensis*, *mesD* and *mesX* were required for methionine biosynthesis in the absence of cobalamin, but *metF* was not.

Finally, we cloned *mesD* and *mesX* from *S. koreensis* into various strains of *E. coli*. *E. coli* encodes both *metE* and *metH*, but in the absence of cobalamin, *metE* is required for methionine biosynthesis. We obtained strains of *E. coli* with *metE* or *metF* deleted from the Keio collection [40]. During aerobic growth in minimal glucose M9 medium, which lacks cobalamin,

the wild-type (parent) strain grows, but neither *ΔmetE* nor *ΔmetF* strains grow (Fig 6). Growth of either *ΔmetE or ΔmetF* strains was rescued when both *mesD* and *mesX* were provided on a plasmid, but not when *mesD* or *mesX* were provided individually (Fig 6). This confirms that MesD is a methionine synthase that requires MesX, but not MetF, for activity.

When we repeated these complementation assays under anaerobic conditions, we found that *mesD* and *mesX* could no longer rescue the growth of *ΔmetE* or *ΔmetF* strains. Given this data and the phylogenetic distribution of MesD/MesX, we conclude that MesD/MesX can only function under aerobic conditions. In principle, this could reflect interactions with the electron transport chain, such as a requirement for ubiquinone. But since neither MesD nor MesX are expected to be membrane proteins [41], we predict that MesD/MesX require molecular oxygen (or perhaps hydrogen peroxide, which is produced by respiring cells) for activity.

### What is the molecular function of MesX?

MesD does not obtain methyl groups from 5-methyl-THF: it lacks the N-terminal folate-binding domain of MetE, and *metF* is not required for MesD's activity. Instead, we predict that MesX provides methyl groups to MesD, either by covalently binding the methyl group, or by forming a methylated substrate that MesD can act on. This would also explain why *mesX* is required for methionine synthesis in *Acinetobacter baylyi* and *Sphingomonas koreensis* and why *mesD* alone is not sufficient for methionine synthesis in *E. coli*.

The requirement for oxygen implies that MesX oxidizes its substrate. Furthermore, MesD and MesX are often encoded near a putative flavin reductase (i.e., ACIAD3522 or Ga0059261_2931); this also suggests that MesX obtains methyl groups by a redox reaction. If MesX is an oxidase, then it cannot obtain methyl groups from other folate compounds such as 5,10-methylene-THF, which would need to be reduced. Because MesD/MesX are found in diverse bacteria and can function in *E. coli*, we predict that MesX obtains methyl groups from central metabolism. As a purely illustrative example, an oxygenase reaction with pyruvate and a reduced flavin could form hydrogen peroxide, glyoxylate, a methyl group attached to a nitrogen or sulfur atom in a side chain of MesX, and oxidized flavin.

### Conclusions

We analyzed four families of core methionine synthases. Based on their distributions, we predicted that MesA obtains methyl groups from the MtrA protein of methanogenesis, while MesB and MesC obtain methyl groups from the CoFeSP protein of the Wood-Ljungdahl pathway. These core methionine synthases may provide a shortcut from central metabolism to methionine: instead of transferring methyl groups from a corrinoid protein to tetrahydrofolate and then to back to another corrinoid protein (namely MetH), it is simpler to transfer the methyl group directly from MtrA or CoFeSP. In contrast to MesA, MesB, and MesC, which are found solely in strictly anaerobic organisms and probably accept methyl groups from corrinoid (cobamide-binding) proteins, MesD is found solely in aerobic organisms, and MesD does not require vitamin B12 or other cobamides as cofactors. We showed that MesD requires MesX (DUF1852) for activity, and that this system functions in *E. coli*, but only under aerobic conditions. We predict that MesD/MesX obtains methyl groups from central metabolism in an oxygen-dependent process. Overall, it appears that diverse bacteria and archaea can synthesize methionine without 5-methyltetrahydrofolate or other methyl pterins as intermediates. We hope that biochemical studies will test our

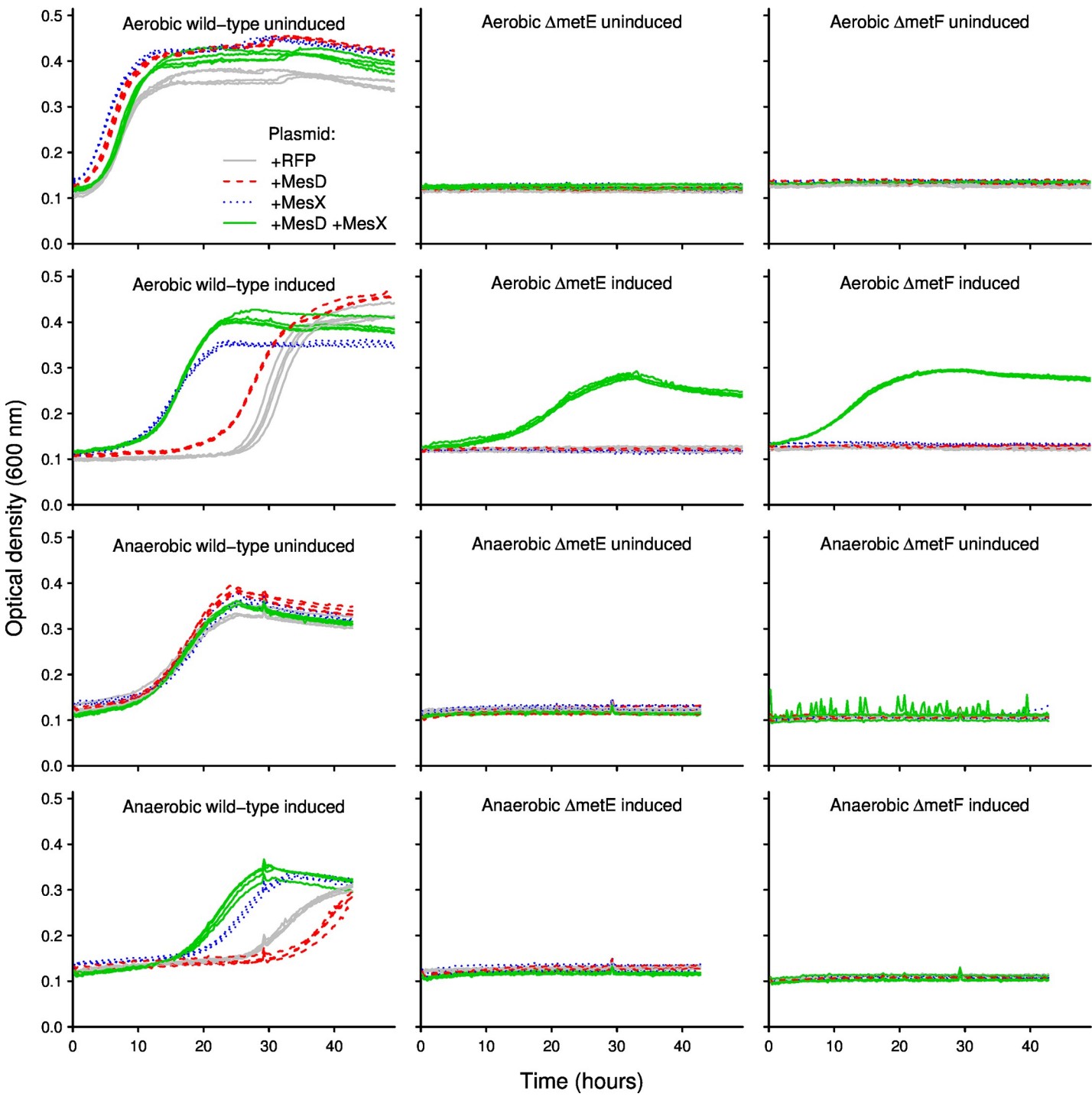

**Fig 6. Complementation assays show that MesD requires MesX and oxygen for activity, but not MetF.** We cloned MesD, MesX, or MesD and MesX together into strains of *E. coli* from the Keio collection [40] and measured growth in minimal glucose M9 medium. A plasmid bearing red fluorescent protein (RFP) was used as a control.

predictions that MesA and MesB use MtrA and CoFeSP (respectively) as methyl donors; that MesC proteins are methionine synthases; that MesX uses a flavin cofactor; and will identify the substrate of MesX.

## Materials and methods

### Literature searches

Literature on MetE and related proteins was retrieved using PaperBLAST [42] and by using PaperBLAST's family search for PF01717 (the catalytic domain).

### Phylogenetic profiling

We downloaded the predicted protein sequences for 321 archaea and 7,694 bacteria from Uni-Prot reference proteomes on October 13, 2020. We searched for MesA, MesB, MesC, MesD, MesX, AcsD/CdhD, AcsC/CdhE, MetE, MetH, MetF, and split MetE proteins. To find homologs of the protein sequences or models (listed below), we used phmmer or hmmsearch from HMMer 3.3.1. For searches with curated models, we used the trusted cutoff (—cut_tc). (Using the gathering cutoff gives identical results.) Otherwise, we used the bit score threshold given below, or else we used E < 0.001. For MesA, we used hits of 174 bits or higher to the characterized protein (METE_METTM). For MesB, we used hits of 173 bits or higher to DET0516 from *Dehalococcoides mccartyi*, and we excluded three non-enzymatic homologs from Chloroflexi. (DET0516 is the MesB protein in *D. mccartyi* 195, and is 99% identical to the characterized MesB protein from *D. mccartyi* CBDB1.) For MesC, we used hits of 38 bits or higher to Q8TUL3. For MesD, we used hits of 560 bits or higher to ACIAD3523. For MesX, we used model PF08908.11 from PFam. For AcsD/CdhD and AcsC/CdhE, we used representatives from *Methanosarcina acetivorans* C2A as queries (ACDD1_METAC and ACDG_METAC, respectively). To identify MetE, we used TIGR01371 from TIGRFam [43]. To identify MetH, we used TIGR02082, but we supplemented these results. During a preliminary analysis of the bacteria that do not contain known forms of methionine synthase, we found that many of them actually contained homologs of the MetH protein from *Thermotoga maritima* (UniProt: Q9WYA5). Biochemical assays have confirmed that the protein from *T. maritima* is a methionine synthase [44], but it scores below the trusted cutoff of TIGR02082. We used protein BLAST with Q9WYA5 as the query to identify additional MetH proteins, with E < 0.001 and at least 80% coverage of the query. The identification of MetF and split MetE proteins are more complex and are described below.

### Searching for methylene-tetrahydrofolate reductase proteins

To identify putative methylene-tetrahydrofolate reductases (MTHFR or MetF), we used model PF02219.1 from PFam. PF02219 is the only domain in characterized MetF proteins, and no other functions for the family have been reported. We also searched for the flavin-independent methylene-tetrahydrofolate reductases [45]. These proteins are distantly related to MetF and do not have statistically significant hits to model PF02219.17. We used the iterative search tool jackhmmer (https://www.ebi.ac.uk/Tools/hmmer/search/jackhmmer) to find homologs of MSMEG_6596 in UniProt reference proteomes. (These preliminary searches were run in May 2020.) The first iteration found 93 hits, all within Actinobacteria; the second iteration found 114 hits; and the third iteration found 541 hits, including 298 proteins with hits to PF02219. We used this third model (at E < 0.001) to identify additional methylene-tetrahydrofolate reductases in all of the UniProt reference genomes (as of October 2020).

For a few genomes of interest, we also used Curated BLAST for Genomes [46] with 1.5.1.20 (the Enzyme Classification number for MetF) to try to find additional candidates. In *Acinetobacter baylyi* and *Sphingomonas koreensis*, we identified the proteins that are annotated as MetF, and we did not identify any proteins in *Dehalococcoides mccartyi* or *Paenarthrobacter aurescens*. But in *A. baylyi*, we also identified ACIAD1783, which is distantly related to

characterized MTHFR proteins (under 30% identity) and lacks the N-terminal part of PF02219. (It aligns to position 70–281 out of 287 in the model.) We are not aware of data about its function, but ACIAD1783 is 39% identical to PP_4638 from *Pseudomonas putida* KT2440 along its full length. 101 fitness experiments for *P. putida* are available in the Fitness Browser (as of October 2020) and PP_4638 has no significant phenotypes. The fitness pattern for *metF* from *P. putida* (PP_4977) is virtually identical to that of the methionine synthase *metH* (PP_2375; r = 0.96), which suggests that PP_4638 did not provide MTHFR activity in these growth conditions. Overall, ACIAD1783 could be a diverged MTHFR protein, but it probably has another function.

## Searching for split MetE proteins

To identify the catalytic component of split MetE, we used phmmer with PF1269 from *Pyrococcus furiosus* (UniProt: METE_PYRFU) as the query, and a threshold of at least 200 bits, which excludes all representatives of MesA, MesB, MesC, MesD, and MetE. We kept only the highest-scoring candidate in each genome. The putative folate-binding component is more divergent and hence more challenging to identify; we used the Pfam model for the N-terminal domain of MetE (Met_synt_1 or PF08267.12) and its trusted cutoff to identify candidates, but if a protein matched the catalytic domain (Met_synt_2 or PF01717.18) as well, then we required that the alignment for Met_synt_1 have the higher bit score. We also required that candidates for either component be at most 400 amino acids long. For comparison, members of the MesA, MesB, MesC, or MesD families were at most 386 amino acids, while the MetE proteins (matching TIGR01371) were at least 676 amino acids. Of 147 genomes with candidates for the catalytic component of split MetE, 107 (73%) contain candidates for the putative folate-binding component; we considered these 107 genomes to contain split MetE proteins.

 Our list of split MetE proteins is incomplete because the putative folate-binding component is difficult to identify. For instance, one of the top-ranking candidates for the catalytic component that was not accompanied by a putative folate-binding component was AMET1_0514 (UniProt: A0A1Y3GEY8) from *Methanonatronarchaeum thermophilum*. However, AMET_0514 is encoded adjacent to a protein that matches both Met_synt_1 and Met_synt_2 with similar scores (AMET1_0515; UniProt:A0A1Y3GF94), and was therefore excluded from the list of candidates for the folate-binding component.

 We did not require that the two components of split MetE be encoded near each other in the genome, and there are a few genomes where they are not nearby (i.e., *Pyrolobus fumarii*). However, across diverse organisms, the two components were almost always adjacent to each other.

 Of the 107 genomes with split MetE, 106 do not encode any of the other known forms of methionine synthase. (The exception was *Haloquadratum walsbyi* DSM 16790, which also encodes MetE.) Of the 106 genomes with split MetE as the sole methionine synthase, 73 are from the order Halobacteria of halophilic archaea and 20 are from the order Thermoprotei of thermophilic archaea. Except for MetE in *H. walsbyi*, we did not identify any other form of methionine synthase in either Halobacteria or Thermoprotei.

## Analysis of MesB and its relatives

To infer a phylogenetic tree of MesB and related proteins, we selected the 88 closest homologs of DET0516 in MicrobesOnline. We removed two truncated homologs and a highly diverged homolog (VIMSS11031200 from *Mahella australiensis* DSM 15567). Using the MicrobesOnline web site, the remaining 85 proteins were aligned using MUSCLE [22], and the alignment was trimmed to relatively-confident columns with Gblocks [47]. We used a minimum block

length of 2 and allowed at most half gaps at any position. A phylogenetic tree was inferred from the trimmed alignment with FastTree 2 and the JTT+CAT model [23]. Fig 3A shows the proteins that are expected to be MesB (given the presence of the Wood-Ljungdahl pathway and functional residues) and their closest neighbors in the tree.

To identify the presence and absence of MetE in these genomes, we used TIGR01371; for MetH, we used COG1410; for MesA, we used BLASTp hits of 180 bits or higher to MTH775 (VIMSS 20772), which is over 90% identical to the characterized protein (METE_METTM); for MesC, we used BLASTp hits of 190 bits or higher to MA0053 (VIMSS 233378) from Methanosarcina acetivorans C2A; for MesD, we used BLASTp hits of 390 hits or higher to ACIAD3523 (VIMSS 590795). For AcsB, we used COG1614 (also known as CdhC). For AcsC, we used COG1456 (also known as CdhE). For AcsD, we used COG2069 (also known as CdhD).

## Aligning functional residues

Substrate-binding residues were determined from PDB:1U1J (MetE from *Arabidopsis thaliana* in complex with zinc and methionine) and PDB:3RPD (MesD from *Shewanella sp.* W3-18-1 in complex with zinc and selenomethionine) using the ligand interaction viewer at rcsb.org and ligplot at PDBsum [48]. We also used the structure-guided aligner MAFFT-DASH [49] to align the C-terminal (catalytic) part of MetE (from *A. thaliana* and from *E. coli*) with MesD from *Shewanella*. Compared to 1U1J, 3RPD has an additional hydrogen bond involving the carboxylate group of selenomethionine and the side chain of Tyr226. Tyr226 is in a MesD-specific insertion that does not align with MetE or with other core methionine synthases, but Tyr226 is conserved within the MesD family.

To identify the corresponding residues in MesA, MesB, and MesC, we used MUSCLE (version 3.7) to align diverse sequences of these families (from MicrobesOnline) to the C-terminal part of MetE proteins. The corresponding residues of MesA from *Methanothermobacter thermoautotrophicus* (UniProt:O26869) were 216,218,240,301 (zinc-binding) and 8,59,174 (homocysteine-binding). The corresponding residues of MesB from *Dehalococcoides mccartyi* 195 (UniProt:Q3Z939) were 214,216,235,311 (zinc-binding) and 15,69,177 (homocysteine-binding). The corresponding residues of MesC from *Methanosarcina acetivorans* (UniProt: Q8TUL3) were 204,206,225,314 (zinc-binding) and 11,59,166 (homocysteine-binding).

To align the representatives of MesA, MesB, MesC, and MesD from UniProt's reference proteomes, we ran MUSCLE separately for each family. To align the members of MetE, we used the model from TIGRFam (TIGR01371) and hmmalign from the HMMer package.

## Mutant fitness assays

Mutant fitness assays used a pool of 260,291 randomly-barcoded transposon mutants of *Sphingomonas koreensis* DSMZ 15582 and were performed and analyzed as described previously [39]. Briefly, the pool was recovered from the freezer in LB media at 30C until it reached log phase. It was then inoculated at $OD_{600}$ = 0.02 into a defined inorganic medium with no amino acids or vitamins and with either 20 mM cellobiose, 5 mM glucose, or 10 mM glutamate as the carbon source. The inorganic base medium contained 0.25 g/L ammonium chloride, 0.1 g/L potassium chloride, 0.6 g/L sodium phosphate monobasic monohydrate, 30 mM PIPES sesquisodium salt, and Wolfe's mineral mix (final concentrations of 0.03 g/L magnesium sulfate heptahydrate, 0.015 g/L nitrilotriacetic acid, 0.01 g/L sodium chloride, 0.005 g/L manganese (II) sulfate monohydrate, 0.001 g/L cobalt chloride hexahydrate, 0.001 g/L zinc sulfate heptahydrate, 0.001 g/L calcium chloride dihydrate, 0.001 g/L iron (II) sulfate heptahydrate, 0.00025 g/L nickel (II) chloride hexahydrate, 0.0002 g/L aluminum potassium sulfate dodecahydrate,

0.0001 g/L copper (II) sulfate pentahydrate, 0.0001 g/L boric acid, 0.0001 g/L sodium molybdate dihydrate, and 0.003 mg/L sodium selenite pentahydrate). Some cultures were supplemented with 0.25 mM L-methionine. These cultures were grown in a 24-well microplate and allowed to reach saturation. The abundance of each strain in each sample was measured by genomic DNA extraction, PCR amplification of barcodes, and sequencing on Illumina HiSeq. Gene fitness values were computed as described previously [50]. Briefly, the fitness of a strain is the normalized log2 ratio of its (relative) abundance in the sample after growth versus in the sample before growth (i.e., at the time of transfer), and the fitness of a gene is the weighted average of the strain fitness values for insertions in that gene. Gene fitness values are normalized so that most values are near zero. All experiments met the previously-published metrics for biological consistency [50].

## Genetic complementation assays

We cloned Ga0059261_2928 (*mesX*), Ga0059261_2929 (*mesD*), or the two genes together into pBbA2c. This vector includes the origin of replication from plasmid p15a, a chloramphenicol resistance gene, the *tetR* regulator, and the inducible P$_{tet}$ promoter [51]. In pBbA2c-RFP (provided by Jay Keasling from the University of California, Berkeley), RFP is downstream of the inducible promoter, and cloning of target genes for overexpression replaces RFP. We used Phusion DNA polymerase and standard cycling conditions for all PCR reactions. Briefly, we linearized pBbA2c-RFP by PCR with oligonucleotides oAD232 and oAD233 and gel-purified this PCR product. (Oligonucleotide sequences are in Table 1.) Ga0059261_2928 (*mesX*) was PCR amplified from total *S. koreensis* genomic DNA with oAD793 and oAD794, Ga0059261_2929 (*mesD*) was amplified with oAD795 and oAD796, and both genes were amplified with oAD793 and oAD796. These inserts were cloned into the linearized and gel-purified pBbA2c using the Gibson assembly mastermix (New England Biolabs) following the manufacturer's instructions. Plasmids with the correct sequence were identified by Sanger sequencing. We then introduced these plasmids (and the pBbA2c-RFP control vector) by electroporation into three strains from the Keio gene deletion collection: the parental or wild-type strain (BW25113), the *metE* gene deletion strain, and the *metF* gene deletion strain [40]. Transformants were selected on LB supplemented with 20 μg/mL chloramphenicol. We performed the complementation growth assays in 96-well microplates using M9 minimal media with 20 mM D-glucose as the carbon source, 10 μg/mL chloramphenicol, and either with inducer (4 nM anhydrotetracycline) or without. (Besides glucose, the M9 medium contained 2 mM magnesium sulfate, 0.1 mM calcium chloride, 12.8 g/L sodium phosphate dibasic heptahydrate, 3 g/L potassium phosphate monobasic, 0.5 g/L sodium chloride, and 1 g/L ammonium chloride.) The microplates were grown in Tecan Infinite F200 readers with constant shaking at 30C and with OD$_{600}$ readings every 15 minutes. For the anaerobic growth curves, we used a plate reader housed in a Coy anaerobic chamber.

**Table 1. Oligonucleotide sequences.**

| Name | Sequence |
|---|---|
| oAD232 | GGATCCAAACTCGAGTAAGGATCT |
| oAD233 | ATGTATATCTCCTTCTTAAAAGATC |
| oAD793 | AGATCTTTTAAGAAGGAGATATACATATGAGCAAGAGCGCATTCACAT |
| oAD794 | AGATCCTTACTCGAGTTTGGATCCTCAGGCGGCGAAGCTGGCC |
| oAD795 | AGATCTTTTAAGAAGGAGATATACATATGAATATAGTGTTGCCCACATC |
| oAD796 | AGATCCTTACTCGAGTTTGGATCCTCAGGCCGAAAGCTCTCGCC |

## Supporting information

**S1 Table. The taxonomic distribution of methionine synthases across UniProt's reference proteomes.**
(PDF)

**S1 Fig. Gene fitness from Sphingomonas koreensis growing in minimal glutamate media with or without methionine.**
(PDF)

## Author Contributions

**Conceptualization:** Morgan N. Price.

**Funding acquisition:** Adam M. Deutschbauer, Adam P. Arkin.

**Investigation:** Morgan N. Price, Adam M. Deutschbauer.

**Resources:** Morgan N. Price, Adam M. Deutschbauer, Adam P. Arkin.

**Software:** Morgan N. Price.

**Supervision:** Adam P. Arkin.

**Writing – original draft:** Morgan N. Price.

**Writing – review & editing:** Adam M. Deutschbauer, Adam P. Arkin.

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
