## [Decision Letter · Decision Letter 0]

18 Dec 2020

Dear Dr Price,

Thank you very much for submitting your Research Article entitled 'Four families of folate-independent methionine synthases' to PLOS Genetics.

The manuscript was fully evaluated at the editorial level and by independent peer reviewers. The reviewers appreciated the attention to an important topic but identified some concerns that we ask you address in a revised manuscript.

We therefore ask you to modify the manuscript according to the review recommendations. Your revisions should address the specific points made by each reviewer.

[LINK]

Yours sincerely,

Josep Casadesús

Section Editor: Prokaryotic Genetics

PLOS Genetics

Reviewer's Responses to Questions

**Comments to the Authors:**

Reviewer #1: This article provides an excellent study of proteins seemingly involved in the last step of methionine biosynthesis - adding a methyl group to the S of homocysteine - a process that typically is understood to require a folate derivative as a methyl donor. While a few articles have shown that the donor in at least a few cases is something other than a folate derivative, Price et al. undertook a study that nicely demonstrates how a comparative genomics study, an exhaustive literature search facilitated by tools the group helped develop, and high-throughput experimental detection of mutant phenotypes such as amino acid auxotrophies, can coalesce into a strong set of conclusions.

The finding is that a number of distinct, although archicturally similar proteins, homologous to MetE over a part of MetE's length, complete methionine biosynthesis by using unexpected donors of the methylgroup added to homocysteine. New names will be awarded, once the findings of this manuscript become available to the public: MesA, MesB, MesC, MesD, MesX, MetH1, MetH2, MetE1, and MetE2. A variety of approaches, including examination of experimental data, was used to support the theory presented for the obligately aerobic role of MesX.

In the Methods, I noted that HMMER3 searches were reported to use --cut_tc as a command line switch. For future work, --cut_ga would be better, if the two are different, since --cut_tc is often set to the lowest score actually observed above the gathering threshold cutoff, rather than lowest score that would be accepted as valid hit - that's how Pfam does it.

The authors spend a bit of their wordcount of refuting the possibility that a certain remote homolog to MetF might actually be MetF. The case is made well, and its demonstration provides a nice illustration of some tools. I support keeping that section of the paper, and mention it to perhaps temper comments that might come from elsewhere suggesting that going even shorter might be better. Presentation seems clear throughout, and length seems appropriate to me. A nice use of bioinformatics methods to help cement an important clarification on how diverse the methods for completing methionine biosynthesis can be, and to help target and interpret the experiments that prove the point.

Reviewer #2: Summary

In the manuscript called “Four families of folate-independent methionine synthases “, the authors perform a comparative sequence analysis of methionine synthase genes across the tree of life. They identify four classes of methionine synthases that differ in their evolutionary history and cofactor use. Focusing on the MesD gene product, they experimentally determine that the protein does not use either of the standard methyl donors (folate or cobalamin) but instead depends on an uncharacterized protein family as well as oxygen.

Major comments

- (Introduction) The authors introduce different types of methionine synthases and outline the key findings of their work. Nevertheless, the introduction misses the formulation of the knowledge gap that this paper is going to fill. Please clearly state the problem or lack of knowledge regarding the core methionine synthases, which is tackled in this work. Please also relate to the problem formulation in the Results and Discussion part if possible.

- This work well presents the differences between known methionine synthases, particularly regarding the cofactor usage, and relates the findings to different domains or phyla. Is it just convergent evolution finding different ways of accomplishing the same objective (synthesis of methionine), or are their implications of picking one cofactor vs another? What would be the advantage of different cofactors being chosen by different evolutionary paths or may the “choice” be caused by differences in the environmental niches? I advise to elaborate on this further in the Discussion part.

- (Conclusion) I ask the authors to comment on future research subjects, experiments etc. to verify the hypotheses stated in this work.

Minor comments

- (Line 99ff) At this point, it is not clear (yet) if the definition of families among the core methionine synthases is a result of this work or was adapted from literature. Please revise.

- Co-occurrence of corrinoid genes with particular methionine synthase genes are the primary evidence for their cofactor use it seems – to what extent has this been experimentally validated?

- (Line 314ff) Can the compatibility of the amino acid change from Glu665 to ASP with zinc binding or coordination be connected to a (or another) zinc-finger motif? Is there a connection between known zinc-finger motifs and the zinc binding residues? Please comment on this, also regarding the mentioned exception in the following paragraph.

- Figure 6: Please clearly state here that the mutants from the Keio collection lack metE and metF, respectively (e.g., include ΔmetE in the headings of the subfigures).

Reviewer #3: This is a perfect manuscript clearly describing the in silico genomic identification and distribution of methionine synthase isozymes across the prokaryotic genomes with complete genomes. The authors have used proper and rigor bioinformatics methods for their analysis and their conclusions are supported by the evidences provided. In addition, the authors has integrated their in silico analyses with experiments on mutant fitness and in vivo genetic complementation to confirm the function of a novel isozyme in a heterologous E.coli host.

There are no major or minor issues with this manuscript, on my opinion. I can only recommend to include additional table (or a figure) that will summarize the distribution of all four identified methionine synthase isozymes across the prokaryotic organisms. it might be a species or higher rank taxa list or a tree on author`s discretion. Figure 3 already shows this distribution at the genome/strain resolution, however only for a subset of taxa that possess the WL pathway. Thus, the proposed summary table may show it at the higher level, e.g. by including the names of respective taxonomic groups (phyla or orders) and representative species that possess the MS isozymes.

**Have all data underlying the figures and results presented in the manuscript been provided?**

Reviewer #1: Yes

Reviewer #2: Yes

Reviewer #3: Yes

PLOS authors have the option to publish the peer review history of their article (what does this mean?). If published, this will include your full peer review and any attached files.

Reviewer #1: No

Reviewer #2: No

Reviewer #3: **Yes: **Dmitry Rodionov

---

## [Editor Report · Decision Letter 1]

5 Jan 2021

Dear Dr Price,

I am pleased to inform you that your manuscript entitled "Four families of folate-independent methionine synthases" has been editorially accepted for publication in PLOS Genetics. Congratulations!

Yours sincerely,

Josep Casadesús

Section Editor: Prokaryotic Genetics

PLOS Genetics

Comments from the reviewers (if applicable):

**Data Deposition**

http://datadryad.org/submit?journalID=pgenetics&manu=PGENETICS-D-20-01730R1

**Press Queries**

---

## [Editor Report · Acceptance letter]

20 Jan 2021

PGENETICS-D-20-01730R1 

Four families of folate-independent methionine synthases 

Dear Dr Price, 

We are pleased to inform you that your manuscript entitled "Four families of folate-independent methionine synthases" has been formally accepted for publication in PLOS Genetics! Your manuscript is now with our production department and you will be notified of the publication date in due course.

With kind regards,

Melanie Wincott

PLOS Genetics

On behalf of:
